# Proteomics Analysis of Andrographolide-Induced Apoptosis via the Regulation of Tumor Suppressor p53 Proteolysis in Cervical Cancer-Derived Human Papillomavirus 16-Positive Cell Lines

**DOI:** 10.3390/ijms22136806

**Published:** 2021-06-24

**Authors:** Pariyakorn Udomwan, Chamsai Pientong, Panwad Tongchai, Ati Burassakarn, Nuchsupha Sunthamala, Sittiruk Roytrakul, Supawadee Suebsasana, Tipaya Ekalaksananan

**Affiliations:** 1Department of Microbiology, Faculty of Medicine, Khon Kaen University, Khon Kaen 40002, Thailand; pariyakorn21535@gmail.com (P.U.); chapie@kku.ac.th (C.P.); panwad1622@gmail.com (P.T.); ati_burass@kkumail.com (A.B.); 2HPV & EBV and Carcinogenesis Research (HEC) Group, Khon Kaen University, Khon Kaen 40002, Thailand; nuchsupha.s@msu.ac.th; 3Department of Biology, Faculty of Science, Mahasarakham University, Mahasarakham 44150, Thailand; 4Functional Ingredients and Food Innovation Research Group, National Center for Genetic Engineering and Biotechnology, National Science and Technology Development Agency, Pathum Thani 12120, Thailand; sittiruk@biotec.or.th; 5Faculty of Pharmacy, Thammasat University (Rangsit campus), Pathum Thani 12120, Thailand; hnungnet@yahoo.com

**Keywords:** human papillomaviruses (HPVs), andrographolide (Androg), tumor suppressor protein p53 (p53), proteomics, cervical cancer

## Abstract

Regardless of the prophylactic vaccine accessibility, persistent infections of high-risk human papillomaviruses (hr-HPVs), recognized as an etiology of cervical cancers, continues to represent a major health problem for the world population. An overexpression of viral early protein 6 (E6) is linked to carcinogenesis. E6 induces anti-apoptosis by degrading tumor suppressor proteins p53 (p53) via E6-E6-associated protein (E6AP)-mediated polyubiquitination. Thus, the restoration of apoptosis by interfering with the E6 function has been proposed as a selective medicinal strategy. This study aimed to determine the activities of andrographolide (Androg) on the disturbance of E6-mediated p53 degradation in cervical cancer cell lines using a proteomic approach. These results demonstrated that Androg could restore the intracellular p53 level, leading to apoptosis-induced cell death in HPV16-positive cervical cancer cell lines, SiHa and CaSki. Mechanistically, the anti-tumor activity of Androg essentially relied on the reduction in host cell proteins, which are associated with ubiquitin-mediated proteolysis pathways, particularly HERC4 and SMURF2. They are gradually suppressed in Androg-treated HPV16-positive cervical cancer cells. Collectively, the restoration of p53 in HPV16-positive cervical cancer cells might be achieved by disruption of E3 ubiquitin ligase activity by Androg, which could be an alternative treatment for HPV-associated epithelial lesions.

## 1. Introduction

High-risk human papillomaviruses (hr-HPVs) are one of the human papillomavirus subsets that can efficiently promote infected-keratinocytes’ transformation to cancers, and they are identified as an etiology of cervical, anogenital, and head and neck cancers [1]. The infection of hr-HPVs signifies a major health problem for the world population, accounting for 5% of cancers globally and affecting 99% of cervical cancers [2].

The intracellular accumulation of two major viral oncoproteins, E6 and E7, plays a functional role in the HPV-induced keratinocyte transformation [3]. The overexpression of E6 and E7 mainly depends on the availability of host-transcription factors [4], e.g., activator protein-1 (AP-1), nuclear factor 1 (NF1), and specificity protein 1 (SP1), that have been reported to activate on the promotor (p97) region of the *E6*/*E7* gene [5]. Notably, E7 directly targets the retinoblastoma protein (pRb), leading to the deregulation of cellular proliferation [6]. Consequently, the E6 oncoprotein potently inhibits several signalings of tumor suppressor pathways, particularly those concerning the p53 cascades, which are important for tumor initiation and cancer maintenance [7].

Recently, despite the anti-HPV vaccine campaigns, these vaccines still fail to achieve any population-therapeutic impact. Moreover, the accessibility of effective vaccinations remains limited in several sections of the world [8,9]. Since no specific anti-HPV drugs are currently offered, new, more specific medications are required, particularly for individuals with no accessibility of vaccination and for infected parties who are at risk of cancer.

Given that E6 enzymatic activity, and its functions, were activated via protein–protein interactions (PPIs) [10], thus, the widely researched E6 roles are the aim of the proteasome-mediated tumor suppressor p53 degradation, which arises in the course of the cellular ubiquitin ligase (E6AP) recruitment [11]. Interestingly, the unceasing activity of E6 is necessitated for the survival of HPV-transformed cells, due to the reactivation of the p53 cascade promoting p53-mediated apoptotic cell death in HPV-positive cancer cells [12]. Then, several anti-cancer therapies, including RNA interference [13], TALEN-based or CRISPR-Cas9-based gene knock-out [14,15], inhibitory peptides, or peptidomimetics [16], and inhibitory drug-like compounds [17], have recently been directly targeted and have disturbed viral E6.

Andrographolide (Androg: C_20_H_30_O_5_) is one of the diterpene lactones purified from *Andrographis paniculata* (Burm. f.) Nees, which is a traditional medicinal plant commonly found in Asia [18,19]. Androg has currently been tested for anti-cancer activities [18,20,21]. Previous in vitro studies demonstrated that Androg suppressed the deregulation of the cell cycle and induced apoptosis in various cancer cell lines including colorectal cancer [22], breast cancer [23], gastric cancer [24], and cervical cancer [25]. The intrinsic and extrinsic apoptotic signaling could be triggered by the Androg compound through the p53-dependent mechanism, activation of reactive oxygen species (ROS) [26], and the topoisomerase II-induced pathway [27]. Androg has been proposed as a promising and potent anti-viral compound. The aforementioned experiments on the anti-viral activities of Androg demonstrated that the medicinal compound prohibited infection and dissemination of the viral pathogens, e.g., HIV [28], Hepatitis C virus (HCV) [29], Influenza A virus [30], Dengue virus (DENV) [31], Chikungunya virus (CHIKV) [32], Hepatitis B virus (HBV) [33], and HPVs [25] to other cells as well as blocking the disease progression by modulating cellular signal transduction. Correspondingly, the previous report by the current authors indicated that Androg and its derivatives prevented the infection of HPV16 pseudo-viruses and significantly inhibited the transcription activity of the long control region (LCR) as well as E6 oncogene expression of HPV16 in transiently transfected C33A cells and in SiHa cells [25]. Moreover, Androg also exhibited anti-cancer activities in HPV-positive cervical cell lines [34]. Currently, the discovery of effective and safe drugs that directly disturb E6-mediated p53 degradation has become practicable, and they are recognized by high-throughput application of liquid chromatography-tandem mass spectrometry (LC-MS/MS) [35,36]. 

Pappa and colleagues [37] analyzed the proteomic profiles of three well-established cervical cancer cell lines—including HeLa, SiHa, and C33A—compared with normal cervical keratinocytes (HCK1T) by LC-MS/MS. Their results suggested the candidate of consistently irregular proteins that might be validated in clinical samples, which could ultimately lead to the disease biomarkers for discovery and drug targets.

While former research by present colleagues revealed the anti-cancer and anti-viral activities of Androg in HPV-positive cervical cancer cell lines, the molecular functions of this compound are still poorly understood. This study aims to investigate the key molecular pathway of Androg-mediated p53 restoration in HPV16-positive cervical cancer cell lines. Through the in vitro model and proteomic analyses in Androg-treated HPV-positive (SiHa and CaSki) and HPV-negative (C33A) cervical cancer cell lines, the down-regulation of the ubiquitin-mediated proteolysis pathway was herein highlighted, particularly the probable E3 ubiquitin-protein ligase HERC4 (HERC4) and E3 ubiquitin-protein ligase SMURF2 (SMURF2) proteins associated with p53 restoration in Androg treated HPV-positive cervical cancer cells, proving a functional role of this compound in the disturbance of HPV E6–p53 interactions.

## 2. Results

### 2.1. Cytotoxicity of Androg on Cervical Cancer Cells

Recently, several active natural compounds have been purified and reported to be cytotoxic on cancer cells, and these have been proposed for clinical cancer phytotherapy. As previously reported [25], Androg (Figure 1A) has been shown to have many biological activities, particularly anti-cancer and anti-viral activities. To investigate the anti-cancer activity of Androg in cervical cancer cells, its cytotoxic activity against three well-known cervical cancer cell lines—SiHa (the integrated form of HPV16 genome, 1–2 copies), CaSki (the mixed form of integrated and episomal of HPV16 genome, 60–600 copies), and C33A—was first determined. The MTT assay proved the cytotoxicity value (CC_50_) of Androg in these cell lines. As shown in the left panel of Figure 1B, 1C and 1D, the effects of Androg on the viability of these cells were promptly observed at 24 h of treatment. The longer incubation of Androg (48 h and 72 h) caused a more than 25% reduction in cell viability at a high dose (160 μM) of Androg treatment. Interestingly, SiHa cells were more sensitive to the treatment than others after treatment with 20, 40, 80, and 160 μM of Androg for 48 h. Cell viability of the SiHa (Figure 1B, *right panel*) and CaSki cells (Figure 1C, *right panel*) was reduced to 50% at a concentration of 85.59 μM and 87.52 μM, respectively, while the CC_50_ of Androg on C33A was 96.05 μM at 48 h (Figure 1D, *right panel*). The sub-cytotoxic concentration (SC) values were calculated as well as 2-fold sub-cytotoxic concentrations (2xSC) of Androg in each cell line (Table 1), and these concentrations were used in subsequent experiments. The results indicated the cytotoxic effect of Androg on cervical cancer cells.

### 2.2. Androg Induced Apoptotic Cell Death in HPV16-Positive Cervical Cell Lines

To assess whether apoptosis is involved in the anti-cancer activities in cervical cancer cells by Androg, a fluorescent EB/AO staining assay was used to determine the morphology of the apoptotic cells in Androg-treated cervical cancer cells. Two fluorescent dyes, labeling different structures of cells [38,39], were used to categorize viable cells and apoptotic cell deaths (Figure 2A). The treatment with SC and 2xSC of Androg for 48 h. significantly induced apoptotic cell deaths in SiHa (6.9 ± 0.51% and 39.9 ± 0.26%) (Figure 2B) and CaSki cell lines (8.5 ± 1.05% and 15.2 ± 0.60%) (Figure 2C) compared to its parentals. Interestingly, these treatments did not show a significant effect on C33A cells (Figure 2D). Thus, these data suggested that Androg specifically caused cell death through the apoptotic process in HPV16-positive cervical cancer cells. To further quantitate the apoptotic cell deaths, flow cytometry was performed in Androg-treated HPV16-positive cell lines. This assay confirmed that apoptosis (early and late) was induced in SiHa cells by Androg treatment at SC (18.7 ± 0.50%) as well as 2xSC (35.9 ± 0.45%) compared to untreated cells (4.5 ± 0.43%), as shown in Figure 2E. Collectively, these results indicated that apoptosis of HPV16-positive cervical cancer cells is associated with Androg treatment.

### 2.3. Proteomic Profiling of Androg-Treated HPV16-Positive Cervical Cells

To gain mechanistic insights into apoptosis of HPV16-positive cervical cancer cells by Androg treatments, proteostatic profiles of the paired Androg-treated cell lines and untreated cells (SiHa-Androg vs. SiHa and CaSki-Androg vs. CaSki) were compared using the LC-MS/MS-based quantitative proteomics study. In this study, two experimental sets, called set SiHa and CaSki, of each set containing two variables of two Androg-treated cell lines and two parental control cell lines were integrated (Figure 3A). In total, 9282 proteins were identified in set SiHa and 8842 proteins in set CaSki. A total of 7739 proteins were observed in 2xSC-treated conditions of the SiHa set while a total of 7601 proteins were discovered in all conditions of Androg treatment in the CaSki set (Figure 3B). Using a 2-fold log change strategy, the expression status of these proteins was next determined. Proteins with higher log 2 ratios than all means were considered to be overexpressed in Androg-treated cells, whereas proteins with log 2 ratios below the mean were estimated to be under-expressed in Androg-treated cells. Accordingly, 2663 proteins were overexpressed in SiHa-Androg and 3342 proteins were overexpressed in the CaSki-Androg in relation to control cells (Figure 3C). In addition, 2627 proteins were downregulated in SiHa-Androg and 2953 CaSki-Androg proteins were downregulated, when compared to its parentals (Figure 3D). Interestingly, 2945 differentially expressed proteins were determined to have been altered in both two Androg-treated SiHa and CaSki, including 766 proteins that were overexpressed and 665 proteins that were down-regulated (Figure 3C–E and Appendix A). Taken together, the proteostasis in cervical cancer-derived HPV16 positive cells were altered by Androg treatment.

### 2.4. Identification of Commonly Regulated Proteins in Androg-Treated HPV16-Positive Cervical Cells

To achieve a functional pathway of the proteomes that are altered by Androg treatments, the 2945 differentially expressed proteins were introduced into GO and KEGG for incorporated network analysis. Several pathways were significantly associated with Androg treatment (*p* < 10^−8^). These include pathways that involve the mitogen-activated protein kinase (MAPK) signaling pathway, mTOR signaling pathway, Wnt signaling pathway, and Ubiquitin mediated proteolysis. (Figure 4A). Importantly, The Kyoto Encyclopedia of Genes and Genomes (KEGG) classification showed that processes including Ubiquitin-mediated proteolysis, autophagy, and mTOR signaling pathway genes were differently expressed in the untreated and Androg-treated HPV positive cervical cancer cell pairs, indicating that Androg treatment might alter the expression of p53-related apoptosis genes (Figure 4B). Therefore, this study was focused on those pathways that were associated with p53 restoration, including ubiquitin-mediated proteolysis. The proteins in this group were consistently down-regulated in the two HPV-positive cell lines compared with their untreated counterparts (Figure 4C). This analysis exhibited that regulation of p53 was markedly involved in Androg-induced apoptosis in HPV16 positive cervical cancer cells.

### 2.5. Androg Altered the Expression of HERC4 and SMURF2 from the Ubiquitin-Mediated Proteolysis Pathway and Restored p53 in HPV16-Positive Cervical Cells

To further evaluate the impact of p53 regulation molecules which responded to Androg treatment, HERC4 and SMURF2 from the ubiquitin-mediated proteolysis pathway were selected and independently analyzed by qRT-PCR. As shown in Figure 4D, the selected molecules demonstrated lower effects in Androg-treated cell lines than the parental cell lines. This information suggests a shared group of proteins that are fundamentally affected by Androg treatment. 

Recent proteomics by this group exhibit the alteration of proteins including the pathway of ubiquitin-mediated proteolysis in HPV16-positive cell lines by Androg that led to the apoptosis event. To test whether the restoration of p53 in HPV16-positive cell lines was affected by Androg treatment, the levels of p53 protein in the treated-cells were next measured. As expected, Androg treatment led to an increase in p53 protein in SiHa (Figure 5A) similar to that of CaSki (Figure 5B) in a dose-dependent manner, indicating the involvement of Androg in the p53 restoration in HPV16-positive cervical cancer cells. The accumulation of p53 protein affects several downstream targets including apoptosis-related proteins. One of the downstream apoptosis proteins, Bax (Bcl-2-associated X protein), has been associated with p53-induced apoptosis through its induction of mitochondrial outer membrane permeabilization (MOMP) that resulted in the leaking of pro-apoptotic factors and the depletion of IAPs (inhibitors of apoptosis proteins), thus introducing intrinsic apoptosis signaling [40,41,42]. In particular, it has been indicated that the p53 protein directly activates Bax transcription, which is driven by the 5′ UTR region of Bax [43]. The transcription level of Bax was then determined in these treated-cell lines. The expression of Bax in HPV16-positive cell lines was highly sensitive to SC of Androg treatment to a similar extent as with the 2xSC, while the expression of the *Bax* gene was not affected in the parental cell lines (Figure 5C,D). The expression of p53 could also be inhibited by HDAC6 directly, as previously described [44]. Collectively, these results indicate that Androg restores the level of p53 protein, thereby promoting downstream apoptosis signaling in HPV16-positive cervical cancer cells.

## 3. Discussion

It is well-known that the infection of hr-HPVs is an etiology of >95% of cervical cancer cases worldwide. Recently, prophylactic vaccines such as Cervarix^™^ and the Gardasil^®^ series have been approved. Despite the campaigns, these vaccines lack significant acceptance. Therefore, medications are still required, particularly for the at-risk individuals who are infected with hr-HPVs or are not vaccinated. Evasion of apoptosis is a hallmark of HPV-associated cervical cancer, and causes ineffective management in patients being treated according to conventional therapeutic strategies [45,46,47]. Moreover, the standard treatments including surgery, radiation, and chemotherapy often fail to cure the cancer, and cause severe systemic toxicity in patients. To this end, researching a phytochemical therapy, a treatment with safe natural-based products, could be a highly feasible alternative. Here, it was demonstrated that the purified andrographolide (Androg), from the Thai medicinal herb *Andrographis paniculata*, restores the expression of p53 via the regulation of the ubiquitin-mediated proteolysis pathway, promoting intrinsic apoptosis in HPV-16 infected cervical cancer.

Subsequently, several active natural compounds have been reported and have exhibited anti-cancer activity when clinically applied for cancer treatment [48]. Herein, this study tested the cytotoxicity of andrographolide against three well-known cervical cancer cell lines, SiHa, CaSki, and C33A. As performed by the standard cell viability assay, MTT, Androg reduced the survival of all cell lines by 50% at a concentration (CC_50_) of 85.59 μM, 87.52 μM, and 96.05 μM. Consistently with the other studies, Androg also caused cancer cell death in leukemia [49], breast cancer [23], prostate cancer [50], esophageal cancer [51], colorectal cancer [22], and oral cancer [52]. Moreover, the report from Alzaharna and colleagues [53] also demonstrated the anti-cervical cancer-derived HPV^+^ activity of andrographolide using HeLa cells as the study model. The anti-cervical cancer activity of Androg using the confocal microscopy by EB/AO staining and flow cytometry with a PI/Annexin V-labelled assay was next studied. Interestingly, Androg enhanced apoptotic cell death in HPV16-positive cells, and specifically exhibited anti-HPV positive cervical cancer activity through the mechanism of apoptosis. It was observed that the treatment of Androg with the optimal concentration at the indicated time point stimulated the EB/AO dye incorporation into HPV-infected cervical cancer cells but was not found in the HPV-negative cell lines. The rearrangement of phosphatidylserine (PS), however, can be used as the marker of early apoptosis. Thus, staining of PI and annexin V was also performed. Flow cytometry revealed that Androg triggered apoptotic programmed cell death in HPV-infected cells, as the percentage of early apoptotic cells increased from 0.8 to 8.5 %, and that of late apoptotic cells increased from 3.7 to 27.4 % as a result of 2xSC treatment. The functional property of Androg as an apoptosis inducer has been also described in the literature in relation to such diseases as colorectal cancer, breast cancer, and gastric cancer. In addition to the PS “flip-flop”, apoptotic cells consist of several unique markers, particularly DNA degradation [54]. A previous study by these authors showed that Androg promoted DNA fragmentation in HPV-positive cells (SiHa) but not in HPV-negative cell lines (C33A) (unpublished data). Therefore, the anti-tumor activity of Androg in HPV16-associated cervical cancer was addressed.

In order to understand the underlying mechanism of Androg inducing HPV16-associated cervical cancer cells’ apoptosis, the proteomic approach (LC-MS/MS) to identify the host proteins that responded to the Androg treatment was used. A total of 2945 proteins were differentially expressed at least 2-fold on the proteomic profile when comparing their expression between the untreated and treated cells. These proteins are involved in several biological functions of the host including the MAPK signaling pathway, mTOR signaling pathway, Wnt signaling pathway, and Ubiquitin mediated proteolysis. Among those pathways identified by LC-MS/MS, the ubiquitin-mediated proteolysis is the key pathway that can degrade the ubiquitinated protein in the cells. Ubiquitination is an important post-translational process that is involved in the pathophysiological mechanisms of many human diseases and cancers. E3 ubiquitin ligases represent a diverse set of enzymes and provide the specificity of the ubiquitination reaction, and have significant roles in many different diseases, especially cancer [55]. In proteomic analysis, it was found that HERC4 and SMURF2 were significantly decreased in the Androg-treated cells. The HERC4 protein is a member of the HECT (homologous to E6AP carboxyl-terminus) domain, which defines a large family of ubiquitin ligases E3 [56]. It has been reported that HECT domains have about 50% similarity to the carboxyl terminal region of E6-associated protein (E6AP) on hr-HPV E6, which possess ubiquitin ligase activity for p53 [57]. E6 is able to bind directly to these target proteins, both independently of E6AP as well as simultaneously to both PDZ domain proteins and E6AP [7]. The interaction of E6 with E6AP directs the ubiquitylation activity of E6AP toward several specific cellular proteins, the most notable of which, with respect to carcinogenesis, is p53 [58]. Moreover, the SMURF2 protein is one of E3 ubiquitin ligases. The SMURF2 protein acts as a molecular editor of DNA topoisomerase IIα, protecting this enzyme from proteasomal degradation and preventing the formation of pathological chromatin bridges, a major cause of chromosomal translocations [59]. In established tumors, SMURF2 could play a role as an oncogene [60]. Emanuelli and colleagues [61] reported that the main alteration in the expression of SMURF2 in prostate and breast tumors is associated with its localization, which corresponds with many reports that the decrease in the nuclear pool of SMURF2 and increase in its cytoplasmic abundance could change the SMURF2′s access to its protein substrates, which include both tumor suppressors and oncogenes. The decrease in the nuclear pool of SMURF2 would diminish its ability to negatively regulate the pro-tumorigenic factors residing in the nucleus (e.g., KLF5, YY1, ID1, SATB1 and others), while increased SMURF2 abundance in the cytoplasm would facilitate the cancer-promoting pathways, including EGFR-induced and KRAS-mediated signaling pathways and, suggestively, the WNT/β-CATENIN pathway (through the degradation of its negative regulators, GSK-3β and AXIN) [61,62,63]. Gupta et.al. [55] reported that the degradation of apoptotic proteins by the ubiquitin-proteasome system (UPS) is essential to the maintenance of cell health, and the deregulation of this process is associated with several diseases including tumors, neurodegenerative disorders, diabetes, and inflammation. Around 100 enzymes of UPS, including ubiquitinates and deubiquitinates (DUB), are associated with the intrinsic, extrinsic, and p53-mediated apoptotic pathways that regulate cell survival or cell death. Based on the proteomic finding, the expression of p53 as well as the downstream pro-apoptotic proteins (Bax) in the treated cells were further measured. The results indicated that Androg restored the p53 protein and the Bax expression level, which could be confirmed by previous studies in which Androg induced p53 protein restoration [25,34].

In the present work, the anti-HPV-associated cervical cancer activity of Androg was validated. Mechanistically, Androg suppressed, in particular, the proteins associated with the ubiquitin-mediated proteolysis pathway, HERC4 and SMURF2, causing p53 restoration, and promoted apoptosis in the infected cells, as shown in Figure 6. Taken together, this study shows that the interference of E6-mediated p53 degradation can be achieved with the medicinal plant compound that promotes cancer cell death in HPV-induced malignant cells and, accordingly, signifies a possible alternative medicinal treatment for HPV-associated epithelial cancers.

## 4. Materials and Methods 

### 4.1. Cell Lines and Cultures

Well-known, established HPV16-positive cervical cancer cell lines, including SiHa and CaSki, and HPV-negative cervical cancer cell lines, including C33A, were used in this study. These cells were cultured in Dulbecco’s modified Eagle’s medium (DMEM; GIBCO, Waltham, MA, USA) supplemented with 10% fetal bovine serum (FBS; GIBCO, Waltham, MA, USA) and antibiotics (100 µg/mL streptomycin and 100 U/mL penicillin) in a humidified atmosphere with 5% CO_2_ at 37 °C.

### 4.2. Andrographolide Compound and CC_50_ Determination

The in-house purified-andrographolide (Androg) compound was used in all experiments. Its structure and pharmacological properties were previously reported by colleagues [25]. The working solution of this compound was freshly prepared, and was dissolved in DMSO to a final concentration. The 50% concentration (CC_50_) of Androg was determined by MTT assay. The 50% cytotoxic concentration (CC_50_) was defined as the compound concentration (µM) required for a reduction in cell viability of 50%, which was calculated by regression analysis. Briefly, all cervical cancer cell lines were seeded in 96-well plates at a density 2 × 10^4^ cells/well. The cells were treated with various concentrations of Androg (0–160 μM) for 48 h. After incubation, 20 µL of MTT solution (5 mg/mL in 1× PBS buffer; AppliChem, New Haven, CT, USA) was added and incubated at 37 °C for 4 h. Then, the mixtures were dissolved with 100 µL of DMSO. The optical density (OD) of each well was measured at 540 nm by a 96-well microplate reader (TECAN, Männedorf, Switzerland). The different concentrations of DMSO (0.3% and 0.6%) and culture media were used as vehicle and blank controls. In addition, a sub-cytotoxic concentration (SC), which is the concentration of Androg that causes up to 15% cell death (CC_15_), was calculated for use in downstream experiments. 

### 4.3. Detection of Apoptosis by Ethidium Bromide/Acridine Orange (EB/AO) Staining and Flow Cytometry

For semi-quantitative analysis, apoptotic cells were detected by the EB/AO solution kit (Merck Millipore, Darmstadt, Germany), at a cell density at 2.5 × 10^4^ in 96 well-plates, and were treated with at SC and 2-fold SC (2xSC) of Androg for 48 h., according to the manufacturer’s protocol, and observed under an Eclipse NiU (confocal) microscope (Nikon Corporation, Tokyo, Japan). The apoptotic cells were randomly counted from three fields in each treatment condition. For the quantitative determination of the Androg-induced apoptosis, a FITC Annexin V Apoptosis Detection Kit II (Invitrogen, Carlsbad, CA, USA) was used. Briefly, a cell density of 1.5 × 10^5^ was seeded onto a 24 well-plate and treated with SC and 2xSC Androg for 48 h., and then, cells were stained with annexin V-FITC and propidium iodide (PI) for 15 min at room temperature (RT) according to the manufacturer’s protocol. Apoptotic cells were then assessed using a BD FACSCanto™ II Cell Analyzer (BD bioscience, Franklin Lakes, NJ, USA). Quantities of 50 µg/mL of Cycloheximide (CHX) and DMSO were used as the chemical-induced apoptosis and blank controls.

### 4.4. Determination of Genes Expression by Quantitative Reverse Transcriptase-PCR (qRT-PCR)

To determine the Androg-altered gene expressions of UPS and apoptosis, the qRT-PCR assay was used. Briefly, total RNA was extracted by TRIzol^TM^ reagent (Invitrogen, Waltham, MA, USA) according to the manufacturer and quantified by NanoDrop™ 2000/2000c Spectrophotometers (Thermo Scientific, Waltham, MA, USA). Approximately, 1 µg of the purified RNA was synthesized to cDNA by RevertAid First Strand cDNA Synthesis Kit (Thermo Scientific, Waltham, MA, USA). The expression of, *HERC4*, *SMURF2* and *BAX* were detected by SYBR^®^ Green Master Mix (Bio-Rad, Hercules, CA, USA) using a CFX96 Touch Real-Time PCR Detection System (Bio-Rad, Hercules, CA, USA). The expression of these genes was normalized with the *GAPDH* gene by the comparative Ct method (2^−ΔΔCt^) [64]. Appendix A shows the specific primers used in this study. 

### 4.5. Measurement of p53 Protein Expression in Androg-Treated Cells by Immunoblotting

Total protein was extracted by TRIzol^TM^ reagent (Invitrogen, Waltham, MA, USA) according to the manufacturer and quantified by the Bradford protein assay using the Bio-Rad Protein Assay Dye Reagent Concentrate (Bio-Rad, Hercules, CA, USA). Approximately, 20 µg of the extracted proteins were separated by 12% SDS-PAGE. The separated proteins were transferred to PVDF membranes Thermo Scientific, Waltham, MA, USA) at 100 V/250 A for 45 min. using the TE77XP Semi-Dry Transfer Units (Hoefer, Holliston, CA, USA). The membranes were then incubated overnight with rabbit anti-p53 monoclonal antibody (Cell signaling technology, Danvers, MA, USA) at 4 °C. After washing with PBS-T buffer (1× PBS buffer with 0.05% Tween-20), the membrane was exposed to the horseradish peroxidase linked goat anti-rabbit monoclonal antibody (Cell signaling technology, Danvers, MA, USA) at room temperature for 1 h. using ImageQuant™ LAS 4000 (GE Healthcare, Piscataway, NJ, USA). The immunoreactive signal was visualized using Amersham™ ECL™ Primers and Western Blotting Detection Reagents (GE Healthcare, Piscataway, NJ, USA). β-actin was used as a control. 

### 4.6. Preparation of Proteomic Samples

After 48 h. treatment with Androg, cells were harvested and added by lysis buffer (7 M urea, 2 M thiourea 2% *w*/*v* of 3-[(3-chloroamidopropyl)-dimethylammonio]-1-propanesulphonate, CHAPS, 0.5% *v*/*v* of IPG buffer; pH 3–10, and 100 mM of dithiothreitol, DTT). Using liquid nitrogen freeze–thaw cycles, total protein was extracted. The protein concentration was determined by the Lowry method. To prepare the proteins for mass spectrometry, tryptic in-gel digestion was performed. As described by Kaewseekhao B et.al. [65], total protein (10 µg) was run with 12.5% polyacrylamide. The gels were dehydrated by acetonitrile (ACN). The sulfhydryl bonds in dehydrated gels were reduced with 10 mM dithiothreitol (DTT) in 10 mM ammonium hydrogen carbonate (NH_4_HCO_3_) at 56 °C for 1 h. The step of alkylation was performed by replacing the DTT with 100 mM iodoacetamide (IAA) in 10 mM NH_4_HCO_3_ and incubated in the dark at RT for 45 min. Gel pieces were then dehydrated with ACN at RT for 5 min. The proteins in the gel were digested by sequencing-grade trypsin (Promega, Mannheim, Germany) at 37 °C overnight. The peptides were extracted from gel pieces with sequential incubations of 50% ACN in 0.1% formic acids at room temperature for 10 min. Then, the mixture was dried at 45 °C for 4 h. The trypsin-digested peptides were protonated with 0.1 % formic acid before LC-MS/MS operation.

### 4.7. Liquid Chromatography-Tandem Mass Spectrometry (LC-MS/MS)

LC-MS/MS was operated on an Ultimate3000 Nano/Capillary LC System (Thermo Scientific, Waltham, MA, USA) and a Hybrid quadrupole Q-Tof impact II™ (Bruker, Billerica, MA, USA) equipped with a Nano-captive spray ion source. Briefly, 20 µL of the sample, with 1 μg of peptides, was enriched on an Acclaim™ PepMap™ 100 C18 HPLC Columns (300 µm i.d. × 5 mm length, 5 µm particle size, 100 Å pore size, Thermo Scientific, Waltham, MA, USA) equilibrated in 2% ACN and 0.1% TFA, for 8 min at 10 μL/min. with an Acclaim PepMap RSLC C18 analytical column, NanoViper (75 µm i.d. × 150 mm length, 2 µm particle size, 100 Å pore size, Thermo Scientific, Waltham, MA, USA). Mobile phase A (0.1% formic acid in water) and Mobile phase B (80% ACN containing 0.1% formic acid) were delivered to the analytical column. Peptides were eluted at 300 nL/min. by gradient the mobile phase B from 5% B to 55% for 30 min. Electrospray ionization was conducted by a CaptiveSpray nano boosted (Bruker, Billerica, MA, USA) in positive mode at 1.6 kV. Inquiry of Mass spectra (MS) and MS/MS spectra was conducted from 150 *m/z* to 2200 *m/z* (Compass 1.9 software, Bruker, Billerica, MA, USA).

### 4.8. Data Processing of Mass Spectrometry (MS)

The identification and quantification of peptides and proteins were conducted using MaxQuant version 1.6.0.13 [66]. The data of MS were investigated against the UniProtKB Human reference database (UP000005640, 71,785 entries; version: March 2017) by Andromeda, the rooted search engine [67]. Fragment ion tolerance was set to 20 ppm and the matching-between-runs option (0.4-min match time window) was enabled. As described by Ruprecht et al. [68], Trypsin was detailed as the proteolytic enzyme with up to two missed cleavage sites authorized. N-terminal protein acetylation and oxidation of methionine were set as variable modifications, while carbamidomethylated cysteine was set as a fixed modification. Exploration outcomes were clarified for a minimum of seven amino acid lengths (1% peptide and protein FDR). If a minimum of two peptides were compared between sample groups, the Label-free protein quantification (LFQ) was calculated. The match between runs feature of MaxQuant was qualified only within experimental replicates. 

### 4.9. Analysis of LC-MS/MS Data

Perseus software version 1.6.0.7 [69] was used for statistical LC-MS/MS data analysis and filtration. To prepare data for PCA analysis, >80% of valid values were filtered and the row median for each protein was subtracted independently. The network proteins-protein interaction was handled and visualized in String version 11 [70] and Cytoscape version 3.7.1 [71]. Term enrichment of gene ontology and the Reactome pathway were computed using both of the Gene Ontology analysis tools (http://geneontology.org/, accessed on 1 May 2021) and the Kyoto Encyclopedia of Genes and Genomes (KEGG) pathway analysis (https://www.genome.jp/kegg/, accessed on 21 April 2021). The variations of pathway activity, over a sample group and in an unsupervised fashion, were estimated through global pathway enrichment analysis using GSVA [72].

### 4.10. Statistical Analysis

To verify the reproducibility, all experiments were conducted in triplicate. Gene expression data from qRT-PCR and apoptosis assays (EB/AO staining and flow cytometric analysis) were subjected to analysis of variance (ANOVA) and Tukey’s test. Data visualization and comparative analysis were conducted in GraphPad Prism 8.0 (GraphPad, California, USA). Results with *p* < 0.05 were considered statistically significant. 

## 5. Conclusions

This study demonstrated that Androg induced apoptosis in HPV16-positive cervical cancer cells and confirmed that p53 restoration was associated with the apoptotic response. This was the leading study to analyze the protein alteration in the cytotoxic effect of Androg on HPV16-positive cervical cancer cells using LC-MS/MS. Our findings indicated that Androg induced apoptosis in HPV16-positive cervical cancer cells through the suppression of the ubiquitin-mediated proteolysis pathway, suggesting that Androg has the potential to be developed as an anti-cancer drug for the treatment of HPV-positive cervical cancer.

## Figures and Tables

**Figure 1 ijms-22-06806-f001:**
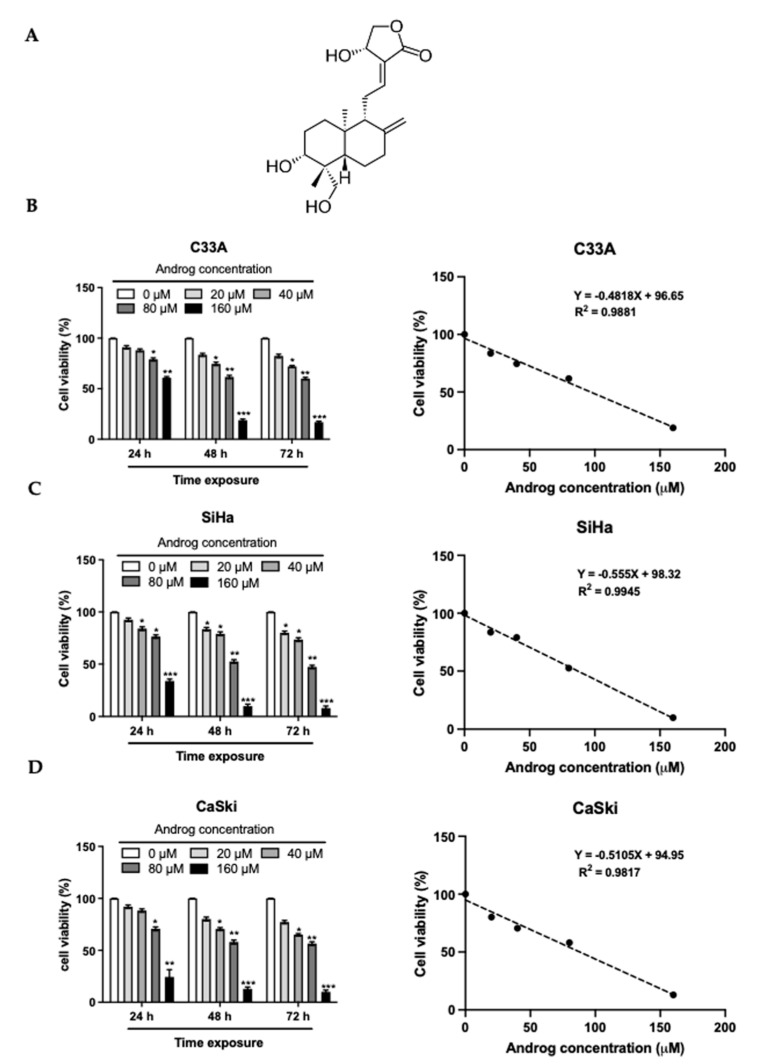
Survival of cervical cancer cell lines after treatment with Androg. (**A**) Chemical structure of Androg. (**B–D**) Measurement of Androg’s cytotoxicity on cervical cancer cell lines in time- and dose-dependent manner (*right panel* = 48-h *exposure*) including, C33A (**B**), SiHa (**C**), and CaSki (**D**), using an MTT assay. * *p* < 0.05 ** *p* < 0.01 *** *p* < 0.001.

**Figure 2 ijms-22-06806-f002:**
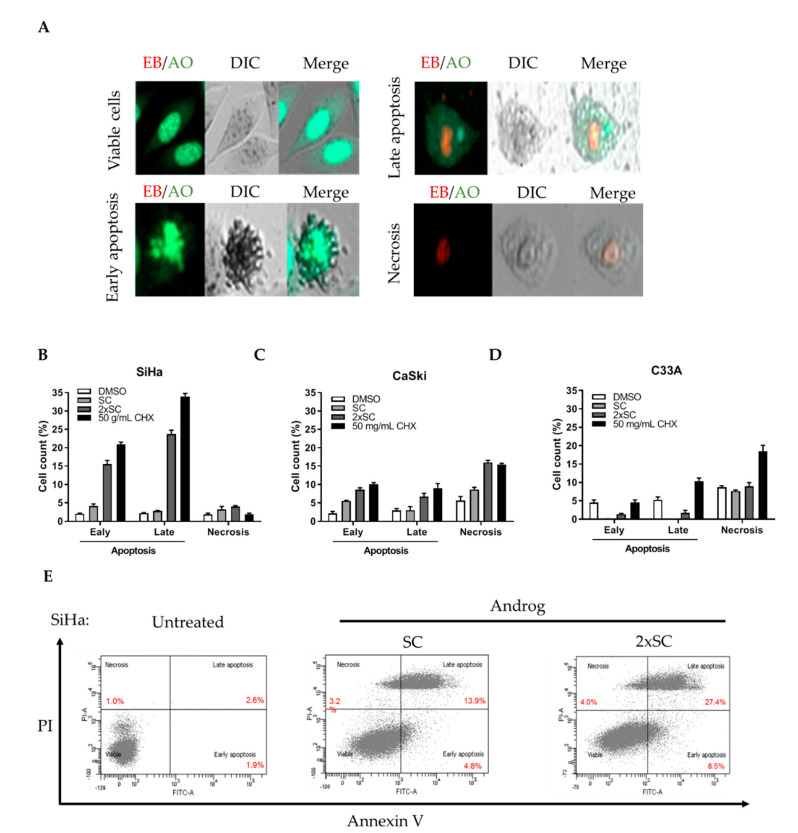
Androg induced apoptosis in HPV16-positive cervical cancer cell lines. (**A**) EB/AO staining of cervical cancer cell lines after treatment with Androg. Red signals (from ethidium bromide, EB) indicated cells that lost their membrane integrity, green signals (from acridine orange, AO) indicated a nucleic acid selective fluorescent cationic dye. (**B**–**D**) Apoptosis in SiHa cells (**B**), CaSki cells (**C**), and C33A cells (**D**) after Androg treatment for 48 h. DMSO and CHX (cycloheximide) were used as vehicle and positive controls for programed cell death. (**E**) The analysis of flow cytometry with Annexin V/PI labeling exhibited the percentages of cells in early and late apoptosis after Androg treatment. SC, sub-cytotoxic concentration; 2xSC, 2-fold of sub-cytotoxic concentration.

**Figure 3 ijms-22-06806-f003:**
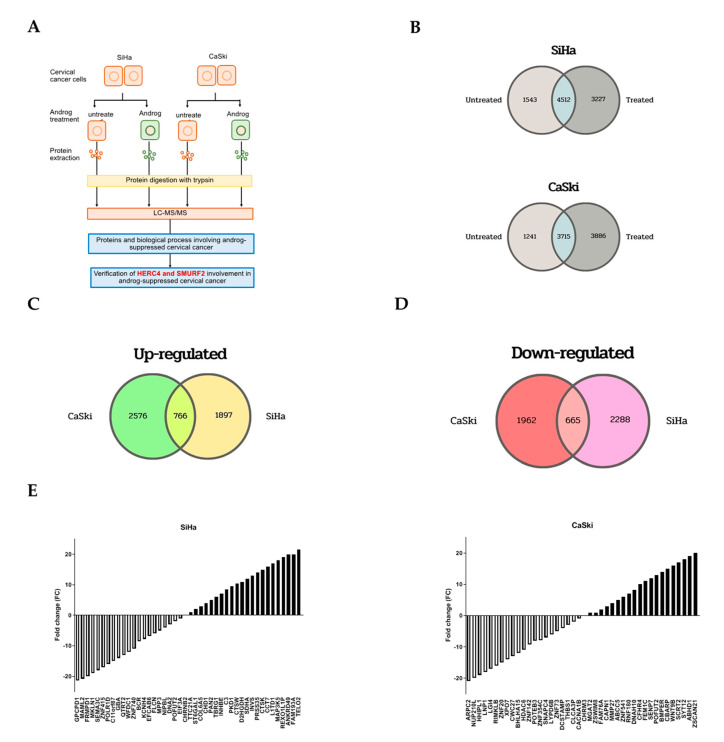
Identification of differentially expressed proteins in Androg-treated HPV16-positive cervical cancer cell lines. (**A**) A schematic diagram exemplified the workflow of proteomic analysis after treatment by Androg with LC-MS/MS. SDS-PAGE was performed using equal amounts of proteins from Androg-treated sublines and their parental, then followed by trypsin digestion. The fractions were injected to LC−MS/MS apparatus. The Proteome Discoverer program, Mascot software, was used as the search engine for the data searching and collection. (**B**) The quantified numbers of proteins in both SiHa (upper) and CaSki (lower) experiments. (**C**,**D**) The numbers of proteins that were determined to be up-regulated (**C**) or down-regulated (**D**) in two Androg sublines, including SiHa, and CaSki cells. Venn diagrams demonstrated the overlap between identified proteins in the two sets. The total number of identified proteins in each set was listed. (**E**) The relative expression levels of differentially expressed proteins in Androg-treated sublines (SiHa = left; CaSki = right) compared with the untreated cells.

**Figure 4 ijms-22-06806-f004:**
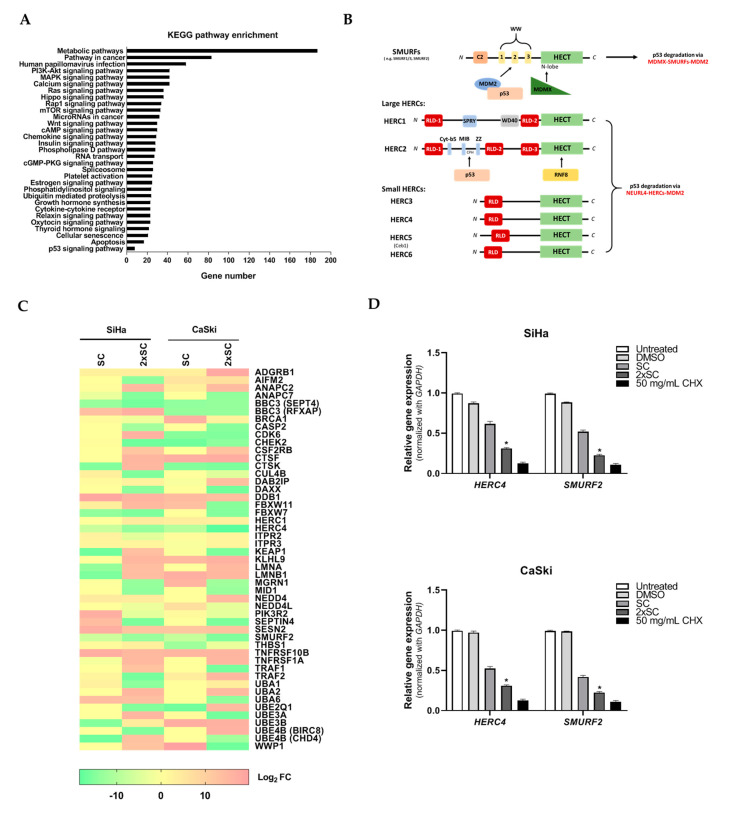
Analysis of the functional pathways of the differentially expressed proteins in Androg-treated HPV16-positive cervical cancer cell lines. (**A**) The functional pathway classifications of the differentially expressed proteins by KEGG platform. (**B**) Identification of p53 protein–protein interaction sites of certain HERC subfamily HECT E3 ligases linked to oncogenesis. (**C**) Overall view of the differential expression of proteins in Androg-treated cells. (**D**) Verification of HERC4 and SMURF2 that were differentially expressed among two paired untreated and Androg-treated sublines by qRT-PCR. * *p* < 0.05.

**Figure 5 ijms-22-06806-f005:**
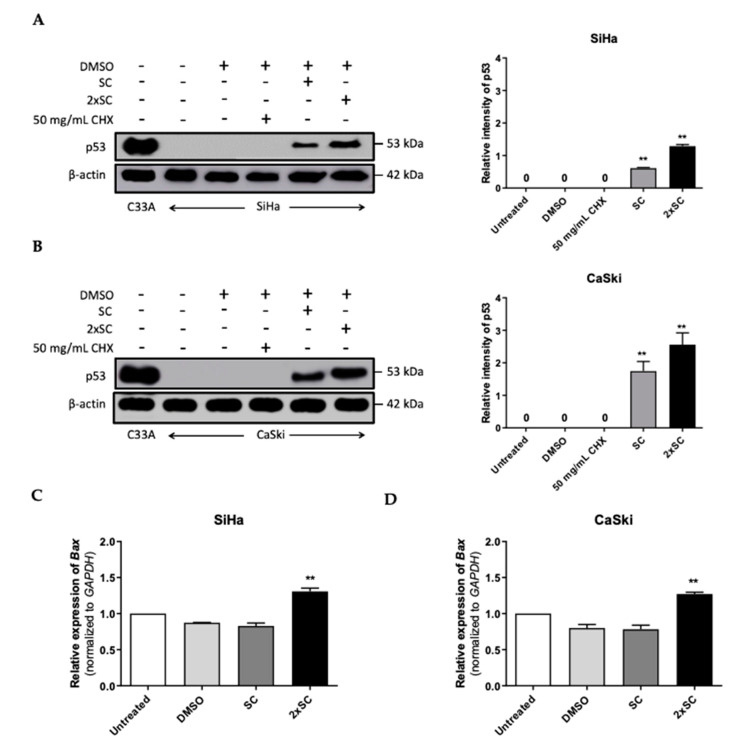
Androg restored p53 expression in HPV16-positive cervical cancer cell lines. (**A**,**B**) The expression of p53 protein in response to Androg treatment was measured by western blotting, as observed in two pairs of HPV16-positive cervical cancer sublines, SiHa (A) and CaSki (B). Column graphs represent the protein intensity of each set performed by ImageJ software. (**C**,**D**) The expression of pro-apoptotic gene, BAX, in response to Androg treatment was measured by qRT-PCR. In each sample, the relative expression of *BAX* was determined after normalization to the *GAPDH* expression level. ** *p* < 0.01.

**Figure 6 ijms-22-06806-f006:**
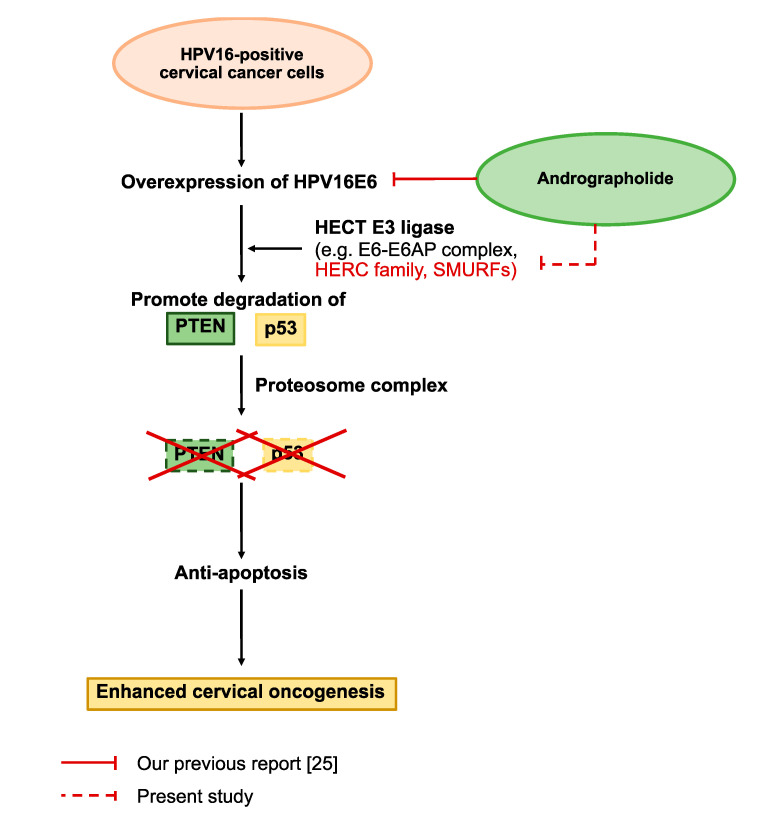
Proposed effects of Androg on inducing apoptosis in HPV16-positive cervical cancer cells. The previous report [25] demonstrated that the expression of E6 was inhibited by Androg in SiHa and CaSki cell lines, leading to programmed cell death in these cell lines. Based on the proteomic profiling of the present study, the apoptosis signaling is also affected by Androg-restored p53 expression via the regulation of UPS-related proteins, including HERC4 and SMURF2.

**Table 1 ijms-22-06806-t001:** The CC50 ^a^, SC ^b^ and 2xSC of Androg on cervical cancer cell lines.

C33A			SiHa			CaSki		
CC_50_ (μM)	SC (μM)	2xSC (μM)	CC_50_ (μM)	SC (μM)	2xSC (μM)	CC_50_ (μM)	SC (μM)	2xSC (μM)
96.05	21.28	42.56	85.59	21.44	42.88	87.52	18.05	36.10

^a^ The 50% cytotoxic concentration (CC_50_) was defined as the compound’s concentration (µM) required for the reduction in cell viability by 50%, which was calculated by regression analysis. ^b^ The sub-cytotoxic concentration (SC) was defined as the compound’s concentration (µM) required for the reduction in cell viability by 15%, which was calculated by regression analysis.

## Data Availability

Not applicable.

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
