# Peer review of "Proteomics Analysis of Andrographolide-Induced Apoptosis via the Regulation of Tumor Suppressor p53 Proteolysis in Cervical Cancer-Derived Human Papillomavirus 16-Positive Cell Lines"

_ijms, 2021, doi:10.3390/ijms22136806_

Round 1

Reviewer 1 Report

The authors present the effects of Andrographolide on inducing apoptosis in HPV16-positive cervical cancer cells. The study first tested the cytotoxicity of andrographolide in cervical cancer cell lines SiHa, CaSki, and C33A and identified CC50 activities in <100 μM range using cell viability assay. Interestingly, Androg enhanced apoptotic cell 254 death in HPV16-positive cells, Androg specifically exhibited anti-HPV positive cervical 255 cancer activity through the mechanism of apoptosis. Mechanistically, based on the proteomic profiling, the anti-tumor activity of Androg essentially relied on the reduction of host cell proteins which are associated with ubiquitin-mediated proteolysis pathways, particularly HERC4 and SMURF2.

Overall, the experiments are rigorous, clearly presented, and compelling and represent an important contribution to the field.

The relatively minor points below should be addressed prior to publication.

Figure-1 represents the activity of three cell lines after 45 h treatment of Androg at different concentrations. Adding the time course experiment for (100uM) here will be useful to understand the potency of the Androg over time.

Reviewer 2 Report

The authors validated the anti-HPV-associated cervical cancer activity of andrographolide natural product. They have showed the interference of the E6-mediated p53 degradation with the andrographolide promoting cancer cell death in HPV-induced malignant cells. The paper is clearly written and recommended to publish in Int. J. Mol. Sci. after the minor text corrections.

  1. I recommend the authors to use either andrographolide or Androg throughout the document for the consistency.
  2. Change the text in line 83 “the previous report by the current authors indicated….” to “in our previous work we indicated…..”
  3. Remove full stop after 48 h in line 140.

Reviewer 3 Report

This is an interesting study about proteomics analysis of andrographolide-induced apoptosis via the regulation of tumor suppressor p53 proteolysis in cervical cancer-derived human papillomavirus 16-positive cell lines. The authors found that the restoration of p53 in HPV16-positive cervical cancer cells might be achieved by disruption of E3 ubiquitin ligase activity by Androg.

The paper is well written. Materials and methods are adequate. Results are well described with graphical summaries.
